# Cardiac 123I-Metaiodobenzylguanidine (MIBG) Scintigraphy in Parkinson’s Disease: A Comprehensive Review

**DOI:** 10.3390/brainsci13101471

**Published:** 2023-10-18

**Authors:** Jamir Pitton Rissardo, Ana Letícia Fornari Caprara

**Affiliations:** 1Neurology Department, Cooper University Hospital, Camden, NJ 08103, USA; 2Medicine Department, Federal University of Santa Maria (UFSM), Santa Maria 97105-900, Brazil; ana.leticia.fornari@gmail.com

**Keywords:** neuroimaging, cardiac imaging, myocardium, 123I, 131I, pre-motor dysfunction, non-motor dysfunction, autonomic dysfunction, parkinsonism, movement disorder

## Abstract

Cardiac sympathetic denervation, as documented on 123I-metaiodobenzylguanidine (MIBG) myocardial scintigraphy, is relatively sensitive and specific for distinguishing Parkinson’s disease (PD) from other neurodegenerative causes of parkinsonism. The present study aims to comprehensively review the literature regarding the use of cardiac MIBG in PD. MIBG is an analog to norepinephrine. They share the same uptake, storage, and release mechanisms. An abnormal result in the cardiac MIBG uptake in individuals with parkinsonism can be an additional criterion for diagnosing PD. However, a normal result of cardiac MIBG in individuals with suspicious parkinsonian syndrome does not exclude the diagnosis of PD. The findings of cardiac MIBG studies contributed to elucidating the pathophysiology of PD. We investigated the sensitivity and specificity of cardiac MIBG scintigraphy in PD. A total of 54 studies with 3114 individuals diagnosed with PD were included. The data were described as means with a Hoehn and Yahr stage of 2.5 and early and delayed registration H/M ratios of 1.70 and 1.51, respectively. The mean cutoff for the early and delayed phases were 1.89 and 1.86. The sensitivity for the early and delayed phases was 0.81 and 0.83, respectively. The specificity for the early and delayed phases were 0.86 and 0.80, respectively.

## 1. Introduction

Parkinson’s disease (PD) is a slowly progressive neurodegenerative disorder affecting more than one million people in the U.S. [1]. The burden of PD on Medicare is estimated to be USD 25.4 billion in indirect medical costs and USD 26.5 billion in indirect and non-medical costs, including an indirect cost of USD 14.2 billion (people with PD and caregiver burden combined), non-medical costs of USD 7.5 billion, and USD 4.8 billion due to disability income received by people with PD [2]. Moreover, in addition to suffering from the classic motor features of PD, people with PD experience significant comorbidities, including increased rates of infections, cardiac and gastrointestinal disorders, and fall-related injuries [3].

The diagnosis of PD requires bradykinesia plus tremor or rigidity. Supportive criteria should be specifically considered in cases where the diagnosis of PD is uncertain or where there are “red flags” or only mild symptoms observed [4]. Sometimes, PD can be challenging to differentiate from other forms of parkinsonism. Drug-induced parkinsonism clinical manifestations can overlap with PD, and drug discontinuation is not always feasible, especially in patients with active neuropsychiatric diseases [5]. In these cases, supportive criteria should be investigated to assist in the differentiation of degenerative and non-degenerative conditions.

In this context, cardiac sympathetic denervation, as documented on 123I-metaiodobenzylguanidine (MIBG) myocardial scintigraphy, is relatively sensitive and specific for distinguishing the causes of parkinsonism. In the last diagnostic criteria for PD by the Movement Disorder Society, cardiac MIBG and olfactory loss were included as supportive criteria [6]. In this way, abnormal cardiac scintigraphy and olfactory testing can increase precision in diagnosing PD [7]. Cardiac MIBG imaging was first approved in 1992 by Japan’s Health and Welfare Ministry to investigate regional denervation of the heart in structural heart diseases [8]. In this context, cardiac MIBG was originally developed to assess postganglionic presynaptic cardiac sympathetic nerve endings in heart disease, including congestive heart failure, ischemic heart disease, and cardiomyopathy [9].

In the mid-1990s, the diagnostic efficacy of cardiac MIBG was further studied in neurological disorders, especially in movement disorders and dementia [10]. In 1994, Hakusui et al. were probably the first to report reduced cardiac MIBG uptake and preserved thallium accumulation in PD [11]. Further studies revealed a significantly lower myocardial MIBG uptake in PD compared with healthy controls. This present study aims to comprehensively review the literature regarding the use of cardiac MIBG in PD with a special focus on procedure techniques, pathophysiological explanations, pre-motor dysfunction, and efficacy to differentiate different forms of parkinsonism.

## 2. Cardiac MIBG Scintigraphy Technique

MIBG, a pharmacologically inactive urea derivative, is an aralkylguanidine analog to norepinephrine. In this way, MIBG and norepinephrine have the same uptake, storage, and release mechanisms [12]. The mechanisms related to these processes are closely associated with norepinephrine metabolism in humans. MIBG binds to the norepinephrine transporter (NET), which is responsible for norepinephrine uptake in presynaptic adrenergic neuron granules and adrenal medullary chromaffin cells [13]. Specifically, MIBG is taken up in cells of neural crest origin, predominantly accumulating in organs with high sympathetic activity, such as the adrenal gland, liver, spleen, heart, and salivary glands [14]. Approximately 90% of MIBG is not metabolized in humans and is mainly excreted unaltered by the kidneys through glomerular filtration. The remaining MIBG is metabolized into m-iodohippuric acid (MIHA) through an unclear pathway [15]. The half-life of MIBG is around 13 h. The elimination is about 50% within the first 24 h and 90% on the fourth day after the injection in patients with normal renal function [16].

There are two types of MIGB uptake mechanisms in neuroendocrine cells (Figure 1). These mechanisms are based on MIBG concentration. In low concentrations of MIBG (uptake type 1), an active pathway dependent on sodium and adenosine triphosphate is chosen. Noteworthy, this is the dominant method of transport of MIBG into cells and is more efficient and specific than type 2 [17]. When inside the cell, MIBG is transported into the storage granules through an energy-dependent transport mechanism via vesicular monoamine transporters 1 and 2. A small portion of MIBG stays in the cytoplasm. When there are high concentrations of MIBG, a pathway called uptake type 2 (extraneuronal uptake) can occur. The mechanism is based on the passive transport of simple diffusion of MIBG into the cells. Interestingly, delayed images are less dependent on uptake type 2 and more accurately reflect cardiac sympathetic nerve activity and pathological findings, but the dose of MIBG may influence these features [18].

The patient should be well hydrated [19], and medication interactions should be checked before the procedure (Table 1). For example, tricyclic antidepressants, antihypertensives, reserpine, calcium agonists, and decongestant drugs such as pseudoephedrine, phenylpropanolamine, and phenylephrine should be held by three to five half-lives [20]. Most current anti-parkinsonian drugs, except monoamine oxidase B inhibitors, do not affect total MIBG myocardial accumulation in PD [21]. Another factor might be that dopamine or its precursor DOPA might act as a competitive inhibitor of NE transporter-mediated MIBG uptake [22]. Experimental studies have shown that MIBG uptake is inhibited by NE, more so by dopamine, and to a lesser extent by DOPA and serotonin in neuroblastoma cells that lack dopamine and serotonin uptake systems [23]. Interestingly, a remarkable decrease in lung uptake in cardiac MIBG imaging may suggest the influence of drugs, so the patient’s medications should be reassessed in the presence of this sign [24]. Structural heart disease (heart failure and ischemic heart disease) and poorly controlled diabetes with small fiber neuropathy are other factors that can interfere with the results of MIBG [25].

It is recommended to slowly administer MIBG, over two minutes, to avoid adrenergic effects [26]. Imaging can be performed minutes to hours after injection of 123I-MIBG. When 131I-MIBG is injected, imaging should be performed 24 to 48 h after the injection [27]. The standard dose of 123I-MIBG varies among countries, such as 111 MBq in Japan, 185 MBq in Europe, and 370 MBq in the USA [28]. Planar images are usually obtained within ten minutes with a matrix of 128 × 128 to 256 × 256 using a gamma camera equipped with a low to medium energy collimator. Single-photon emission computed tomography (SPECT) can provide images that can be rotated 180 to 360 degrees, in which polar maps can be applied and may reflect better anatomical localization of possible lesions. However, the total time to obtain SPECT images is increased to thirty minutes with an anger camera and ten minutes with a cardiac cadmium–zinc–telluride detector [29].

Cardiac images are usually obtained between 10 and 30 min, known as early registration, and after 3 to 4 h, known as delayed registration, of the radiolabeled injection [30]. Noteworthy, the potential role of immediate acquisition of images within 3–5 min after MIBG injection in parkinsonism has been investigated. Frantellizzi et al. found that the heart-to-mediastinum ratio (H/M ratio) mean values were 1.58 ± 0.22 for immediate (5 min), 1.61 ± 0.26 for early (15 min), and 1.59 ± 0.37 for late (240 min) acquisitions. Also, they proposed immediate registration to differentiate cardiac MIB uptake from structural heart diseases, in which the acquisition should be obtained at 240 min based on the pathophysiology of heart failure [31]. On the other hand, Chun et al. calculated the efficacy of cardiac MIBG uptake and found abnormal results in individuals with PD at 15 min, 1 h, 2 h, 3 h, and 4 h post-injection of MIBG. The authors recommended that 4-hour-delayed imaging is the best diagnostic performance [32].

Early registration is associated with the influx of MIBG into extraneural spaces, like the myocardial tissue. In this context, the early phase can demonstrate the integrity and distribution of the presynaptic sympathetic system and the density of the presynaptic cardiac sympathetic nerve endings. Delayed registration reveals the neuronal uptake of MIBG and correlates to sympathetic nerve terminals’ functional status and tone [33].

The H/M ratio measurement in the early and delayed phases and the washout period are the most important parameters obtained in cardiac MIB scintigraphy [34]. During registration, MIBG uptake can be quantified by the MIBG uptake in the heart (counts per pixel in the myocardium) versus the MIBG uptake in the mediastinum (counts per pixel in the upper mediastinum), where the relationship between these two variables is the H/M ratio. Another possible method to evaluate the autonomic function is the washout period of MIBG. The MIBG washout rate (WR), corrected for decay, is calculated using the formula: WR = [(early(H/M) − delayed(H/M))/early(H/M)] × 100. The washout rate is the ratio of cardiac uptake between early and delayed scans, which may be associated with catecholamine release and sympathetic tone. In this way, a long washout period may indicate worsening cardiac function, and the washout rate can be a degree the autonomic dysfunction [34,35].

Post-imaging adverse events were rarely reported but may include tachycardia, pallor, vomiting, and abdominal pain. Thyroid blocking with potassium iodide is recommended for MIBG scintigraphy due to radionuclide 123I. Potassium iodide competitively inhibits radioiodine uptake, preventing excessive radioiodine levels in the thyroid and minimizing the risk of thyroid complications. Potassium iodide or Lugol’s solution containing 100 mg iodide, weight-adjusted, should be given one hour before the injection [36].

The most common area of study with 123I-MIBG is the cardiac region. However, other organs are useful in investigating sympathetic activity, such as the liver, lungs, muscle tissue, and parotid glands [37].

Haqparwar et al. investigated MIBG uptake in the major salivary glands as a potential biomarker for diagnosing PD. The authors found that PD patients showed a significantly lower MIBG uptake in the parotid and submandibular glands than the controls (*p* < 0.0001). This finding could be explained by pathophysiological mechanisms involving extracranial alpha-synuclein of the superior cervical ganglion, which is independent of cranial nigrostriatal alpha-synuclein. It is worth mentioning that the parotid MIBG uptake in PD patients did not correlate with clinical severity or disease duration [38]. Some authors believe that the study of cardiac MIBG imaging should be performed with a salivary gland assessment. Combining major salivary gland and myocardial scintigraphy results in the early PD period showed a significant improvement in the diagnostic value of MIBG imaging [39].

A study revealed that reduced thyroid MIBG correlated with PD but not diabetic autonomic neuropathy [40]. This finding is important because diabetes mellitus significantly limits cardiac MIBG performance due to the higher number of false-positive results associated with this condition. However, one important drawback of directly studying thyroid tissue is the thyroid damage related to the radionuclide 123I in [123I]-MIBG [41]. Nevertheless, further studies with other organs and radionuclides in patients with PD should be performed to improve the understanding of the pathophysiologic mechanisms underlying neurodegeneration in this condition.

## 3. Parkinson’s Disease and Cardiac 123I-MIBG

### 3.1. Physiological and Anatomical Changes in the Cardiovascular System of Parkinson’s Disease

Spinal cells are responsible for the sympathetic innervation of the heart. These cells originate in the upper 3–4 thoracic segments of the spinal cord. The first synapses form in the stellate (cervicothoracic) and thoracic sympathetic ganglia. Postganglionic noradrenergic sympathetic fibers accompany the blood vessels to the heart and enter into the myocardium. Parasympathetic fibers relay at ganglia located directly on the heart and short postsynaptic fibers. The main areas innervated by the parasympathetic nervous system are the atrial muscle and the sinoatrial and atrioventricular nodes. But, the ventricular myocardium is sparsely innervated by vagal efferents [42].

Lewy bodies are intra-cytoplasmic eosinophilic inclusions with a hyaline core and a pale halo mainly composed of aggregated α-synuclein. In the early stages of PD, Lewy bodies accumulate significantly in structures of the lower brainstem and the olfactory system [43]. At this stage, also known as Braak stage 1, the dorsal motor nucleus of the vagus nerve in the medulla oblongata and anterior olfactory nucleus are affected. The involvement of the vagus nerve can partially explain the denervation of some cardiac structures [44].

Mitsui et al. performed a neuropathological study on a 71-year-old male with a diagnosis of PD who had a cardiac MIBG performed one year before the autopsy [45]. The patient had a low MIBG uptake in the early and delayed phases. Some neuropathological findings on this individual can contribute to understanding cardiac involvement in the pathophysiology of PD. First, Lewy bodies were present in the intermediolateral column of the thoracic spinal cord and sympathetic ganglia. Second, immunohistochemistry with anti-phosphorylated α-synuclein antibody was positive on axons in the thoracic ventral roots, sympathetic trunk, and cardiac plexus. Third, total loss of anti-tyrosine hydroxyls immunoreactivity and a marked decrease in axons were observed in the cardiac plexus. Fourth, no valvular, coronary, or myocardial abnormality was observed in the gross and microscopic anatomy of the heart.

The findings of cardiac MIBG studies correlated with pathology reports are important for understanding the progression and involvement of the cardiac system in PD [46]. The uptake reduction in MIBG in the heart was found to be associated with denervation of the heart, leading to hyperdynamic cardiac contractility in response to adrenergic responses related to beta 1 [47]. Also, patients with reduced MIBG uptake may have reduced cardiac contractility during exercise, suggesting an impaired response to exercise capacity [48]. A possible explanation for this finding is the sympathetic denervation observed in individuals with PD.

Takahashi et al. studied the extent of cardiac sympathetic denervation in cases of Lewy body disease confirmed using autopsy and the relationship between residual sympathetic nerve preservation and cardiac MIBG uptake in life. The authors found a quantitative correlation between cardiac imaging and the loss of sympathetic axon loss in cardiac tissue samples [49].

The efficiency of vesicular sequestration in individuals with PD can provide significant information on neuronal death. In this context, cardiac MIBG cannot quantify the functional abnormality, but the accelerated loss (“washout”) of MIBG might provide a biomarker of increased sympathetically mediated exocytosis [50].

The morphometric study of the brain can add important information regarding indirect functionality and neuronal damage. Kikuchi et al. evaluated voxel-based morphometry and diffusion tensor imaging in relationship to cardiac MIBG in individuals with PD. They found that individuals with an abnormal H/M ratio compared with those with normal had reduced brain volume at the right inferior frontal gyrus and lower fractional anisotropy at the left anterior thalamic radiation, the left inferior fronto-occipital fasciculus, the left superior longitudinal fasciculus, and the left uncinate fasciculus [51].

Park et al. studied cardiac MIBG, 18F-FP-CIT, and striatal dopamine transporter imaging in individuals with PD [52]. Striatal 18F-FP-CIT uptake did not correlate with plasma α-synuclein levels. In individuals with early PD, plasma α-synuclein levels correlated with cardiac sympathetic denervation found in the cardiac MIBG (*p* = 0.01) but not with nigrostriatal degeneration (*p* = 0.61) in a multivariate analysis. This may suggest that plasma α-synuclein levels more readily reflect the peripheral accumulation of Lewy bodies than central. Furthermore, it may suggest that central and peripheral deposition of Lewy bodies could have, in fact, two different pathological pathways [53].

One of the possible pathophysiological mechanisms hypothesized in PD is related to the serotoninergic pathway [54]. Brain postmortem analysis of PD patients showed that the density of serotonergic neurons and the concentrations of serotonin and its metabolites were decreased in the raphe area of PD patients [55]. Some authors found a link between the central serotonergic and cardiac sympathetic systems. The cerebrospinal fluid 5-hydroxyindole acetic acid concentration correlated with a delayed H/M ratio (r = 0.458, *p* < 0.05) and the washout rate (r = −0.642, *p* < 0.01). Therefore, these results can support the concurrent progression of central serotonergic and cardiac sympathetic dysfunction in individuals with PD [56].

### 3.2. Cardiac MIBG Scintigraphy as a Supporting Criteria for PD

For many years, the main diagnostic criteria for PD were only based on clinical cardinal features. Hughes et al. performed a clinic-pathological study with one hundred individuals with PD. The U.K. PD Society Brain Bank criteria for PD diagnosis had a specificity of 93%, meaning that 7% of the individuals with PD were misdiagnosed [57]. To address this concern, the Movement Disorder developed diagnostic criteria for PD improving the specificity with a balanced sensitivity. These new criteria included two levels of certainty, supportive studies, absolute exclusion criteria, and red flags. One supportive criterion in diagnosing PD is either olfactory loss or cardiac sympathetic denervation on MIBG scintigraphy [6].

Figure 2 summarizes the cardiac MIBG results found in different studies. Most studies consider a normal H/M ratio as above two. However, several factors directly and indirectly related to the patient and the collimator can influence these results. Also, significant differences exist in the early and delayed phases of registering the H/M ratio. Interestingly, some clinical features in patients with PD are statistically associated with early and delayed cardiac MIBG uptakes. These symptoms are rapid eye movement sleep behavior disorder, urinary disturbances, rigidity, and constipation. Moreover, tremor is only associated with delayed cardiac MIBG uptake [58].

The most common form of imaging acquisition of cardiac MIBG is two-dimensional planar images. In this context, three-dimensional imaging using single-photon emission tomography (SPECT) may provide a more complete understanding of regional cardiac sympathetic innervation with an assessment of the polar model (Figure 3). The conventional 17-segment/five-point model used for SPECT myocardial perfusion imaging is usually used [59]. Studying cardiac segments with SPECT, in addition to the H/M ratio, is important for patients with particular anatomical deformities. Also, they may provide additional information to develop scores with lower variability between patients and be less dependent on the quality of the imaging acquired. The literature about the SPECT of cardiac segments involved in differentiating parkinsonian syndromes is scarce. However, there are many studies in patients with structural heart conditions.

Kwon et al. showed that the summed defect score and defect scores in the anteroseptal and inferior regions are statistically significant in differentiating PD from essential tremors [60]. Furthermore, Courbon et al. described a prominent regional reduction in MIBG uptake in the apex region in patients with PD compared with multiple systemic atrophy [61]. The results of both studies suggest that cardiac denervation might affect mainly the anteroseptal region, including the apex, in PD patients, which is the main distribution of sympathetic fibers in the heart.

The inferior wall of the ventricles was also associated with a decreased uptake of MIBG in individuals with PD [60]. However, this area should be cautiously analyzed. The inferior wall is related to many artifacts because images are rendered to a planar surface on a different axis. Also, the liver is one of the areas with high uptake of MIBG, causing difficulties in the reconstruction due to artifacts. In an elderly population, studies showed a strong association between decreased uptake of MIBG in the inferior wall region and the aging effect [62].

The cardiac MIBG SPECT uptake distribution can be homogeneous, non-homogeneous, or absent. In the non-homogenous distribution, interesting patterns of uptake can be observed. The uptake of the apex is commonly observed in PD and multiple system atrophy, but apparently, uptake can only be complete in PD. Also, the dysfunction of multiple system atrophy usually affects the left circumflex artery territory more than the other coronary territories [63]. Interestingly, a similar non-homogeneous pattern has been found with 6-[18F]- fluorodopamine [64] and [11C]-hydroxyephedrine [65] PET scanning in patients with PD.

### 3.3. Cardiac MIBG Scintigraphy in Parkinson’s Disease

In the 1990s, it was observed that patients with PD disclosed a significantly lower myocardial MIBG uptake than healthy controls. This was especially observed in the H/M ratio, where patients with PD had a significantly lower H/M ratio than healthy controls [66]. Reduced myocardial MIBG uptake reflects sympathetic myocardial degeneration in PD and the results from a Lewy body type degeneration of the cardiac plexus. Noteworthy, there are significant discrepancies in the literature results regarding the correlation between MIBG and age, sex, years of the disease, and Hoehn and Yahr stage.

Patients with a mild Hoehn Yahr stage or short-duration disease, compared with healthy individuals, will have reduced myocardial MIBG uptake. Interestingly, one study investigated individuals with different Hoehn and Yahr stages, and the authors observed that in patients with more advanced PD (Hoehn and Yahr stages II–V), compared with early PD (Hoehn and Yahr stages) and healthy individuals, myocardial MIBG uptake will have high sensitivity and specificity [67].

Table 2 shows the sensitivity and specificity of cardiac MIBG scintigraphy in PD to differentiate it from other forms of parkinsonism. Fifty-four studies involved 3114 individuals with PD with a mean Hoehn and Yahr (H&Y) stage of 2.5. The mean early and delayed registration H/M ratios were 1.70 and 1.51, respectively. The mean cutoff for the early and delayed phases were 1.89 and 1.86. The sensitivity for the early and delayed phases was 0.81 and 0.83, respectively. The specificity for the early and delayed phases were 0.86 and 0.80, respectively.

### 3.4. Autonomic Function in PD and Cardiac MIBG

The investigation of autonomic function in the clinical setting is based on measuring blood pressure, heart rate, variability in the heart rate, and the effect of different body positions on blood pressure. The vagus nerve influences parasympathetic activity in the heart, which has a fast response with higher frequency modulation. On the other hand, sympathetic activity in the heart is associated with the paravertebral ganglia and has a slow response with lower frequency modulation. There are several tests to investigate autonomic function, including orthostatic reaction, deep breathing, and the Valsalva maneuver [116]. Cardiac MIBG is a method that can particularly assess sympathetic myocardial function.

The assessment of autonomic function with different methods and cardiac MIBG in individuals with PD has contradictory results. Nakahara et al. assessed the correlation between heart rate variability analysis and cardiac MIBG uptake in patients with PD and found no association between these two variables [117]. In this context, some authors observed a significant correlation between cardiac MIBG uptake and sympathetic and parasympathetic function [33]. However, other authors found no statistically significant correlation [118]. Berganzo et al. studied the relationship between Scales for Outcomes in PD–Autonomic chapter (SCOPA-AUT) scores and cardiac MIBG uptake. The severity of dysautonomia measured using SCOPA-AUT was not correlated with clinical severity, time since onset, or the early and delayed registrations of the H/M ratio. In patients with PD, the only variable associated with a delayed H/M ratio was the age at the onset of the disease [119].

Manabe et al. found an interesting correlation between systematic blood pressure and cardiac MIBG uptake [120]. They monitored the circadian blood pressure patterns of 37 patients with PD. The authors found that a nocturnal percentage decline in arterial blood pressure was associated with the H/M ratio on early and delayed images (*p* < 0.01). In this context, Kim et al. discovered that orthostatic hypotension was closely associated with cardiac sympathetic denervation observed on cardiac MIBG in patients with early and mild PD [121]. This is interesting because ambulatory blood pressure can be easily accessed in clinical practice. Also, the measurement of systematic blood pressure can be used as a primary step for assessing the significance of further workup. In cases without neurogenic orthostatic hypotension, baroreflex sensitivity and low-frequency diastolic blood pressure are the best predictors of cardiac sympathetic denervation at rest [122].

Individuals with PD presenting with syncope/presyncope have more common dysfunction of the cardiovascular autonomic system. In this way, some authors proposed that cardiac MIBG should be used to help identify patients with an elevated risk of syncope episodes and that the results should be used to choose the best management [123]. Other studies found an association between the body mass index and the development of cardiac MIBG uptake abnormality in individuals with PD [124]. However, these data should be cautiously analyzed due to the possible influence of metabolic diseases in the autonomic system, leading to abnormal cardiac MIBG results.

The Valsalva maneuver is divided into four phases based on systolic blood pressure and heart rate. The last phase is associated with an overshoot of blood pressure due to activation of the sympathetic system [125]. Also, the overshoot of blood pressure leads to baroreflex stimulation, leading to bradycardia and the return of blood pressure to the baseline. The fourth phase is directly related to sympathetic function, so in patients with PD, the most affected phase will be the last. One study found that cardiac MIBG uptake was moderate (r = 0.648, *p* = 0.0003) and associated with a reduction in the overshoot of the fourth phase of Valsalva maneuvers in individuals with PD [126]. This association could reflect postganglionic sympathetic noradrenergic impairment but also central baroreflex-sympathoneural failure. Other authors found an association between cardiac MIBG uptake and the coefficient of variation for intervals in resting and deep breathing, suggesting parasympathetic dysfunction [127]. Therefore, sleep behavior disorder may be not directly associated with cardiac MIBG; instead, it is autonomic dysfunction that plays a role in the study of both sleep behavior disorder and cardiac MIBG.

Autonomic dysfunction may be a confounding factor in many studies investigating specific clinical manifestations of PD with cardiac MIBG. For example, sleep behavior disorder in PD was already considered an independent risk factor for abnormal cardiac MIBG uptake. However, this could be a confounding factor between autonomic dysfunction and PD progression without peripheral signs. Therefore, sleep behavior disorder maybe not be directly associated with cardiac MIBG. Instead, it is autonomic dysfunction that plays a role in the study of both sleep behavior disorder and cardiac MIBG.

### 3.5. Parkinson’s Disease Subtypes and Cardiac MIBG

Three main types of PD are characterized by tremor-dominant, akinetic-rigid form, and mixed condition. There are significant clinical differences among these historical subtypes. Chung et al. revealed that individuals with tremor-dominant PD usually have normal cardiac MIBG uptake, but the akinetic-rigid and mixed subtypes have lower levels of cardiac MIBG uptake. No statistical difference was observed between akinetic-rigid and mixed subtypes regarding cardiac scintigraphy [128]. Noteworthy, Chiaravalloti et al. presented opposite findings, where the cardiac sympathetic system was more severely impaired in the tremor-dominant subtype than in the akinetic-rigid subtype [129]. But the Chiaravalloti et al. cohort of individuals with tremor-dominant was older, which can be associated with more confounding variables such as aging and possible comorbidities influencing the autonomic nervous system.

Some indirect assumptions can be made from the findings of Chung et al. In this way, the tremor-dominant subtype may be a less severe disease with minimal progression to the periphery. Interestingly, patients presenting with the predominant akinetic-rigid form have a more severe motor impairment, more severe disabilities, and more commonly present non-motor symptoms [130].

### 3.6. Genetic Causes of Parkinson’s Disease

The neuropathological and clinical manifestation findings vary among the different forms of genetic parkinsonism. Parkinson’s disease-1 (PARK1) carriers usually have an aggressive Lewy body pathology [131]. On the other hand, in patients with leucine-rich repeat kinase 2 (LRRK2) gene mutation (PARK8), variable degrees of Lewy body pathology are observed [132]. Interestingly, some patients, mainly those with PARK2, are considered almost free of Lewy body lesions. Therefore, the study of cardiac MIBG in these pathologies can significantly contribute to understanding the clinical manifestations and spectrum of these conditions.

The results found in the literature regarding genetic causes of PD are controversial. Two studies found that individuals with hereditary PD, compared with idiopathic PD, had normal or less impaired cardiac MIBG uptake. However, the studies did not perform a subgroup analysis of the different genetic forms of PD with cardiac MIBG uptake [133,134]. Gabilondo et al. studied 194 patients with suspected synucleinopathy or atypical parkinsonism, in which 34 individuals had a genetic diagnosis of PD. The authors found a significantly reduced uptake for individuals with PARK1 but not for PARK2 or PARK 8 [108]. Tijero et al. found similar results: cardiac MIBG abnormalities were less common in PARK2 mutation carriers than in patients with idiopathic PD [135].

Ruiz Martínez et al. investigated olfactory dysfunction and the changes in cardiac MIBG uptake in patients with PD carrying R1441G and G2019S mutations in LRRK2 [136]. The authors found that olfactory and cardiac impairments are less prevalent when PD is associated with mutations in LRRK2. In early registration, the difference was statistically significant, where LRRK2 noncarriers had a H/M of 1.49 ± 0.28 and LRRK2 carriers had 1.75 ± 0.38 [136]. Interestingly, Valldeoriola et al. [137] found similar results as the study of Ruiz Martínez et al. [136]. Also, Valldeoriola et al. revealed a moderate relationship between olfactory testing and the H/M ratio on early and late cardiac MIBG uptake (early: r = 0.62 *p* < 0.02; late: r = 0.68 *p* = 0.01) [137].

Patients with PD with glucocerebrosidase gene mutations are known to show more rapid clinical progression than sporadic PD without glucocerebrosidase gene mutation [138]. Kim et al. studied patients with PD with and without this mutation. They found that cardiac sympathetic denervation and non-motor symptoms (orthostatic hypotension, rapid eye movement sleep behavior disorder) were more common in individuals with glucocerebrosidase gene mutations. Also, the delayed H/M ratio between the two groups was significant (*p* = 0.043), in which individuals with the mutation had 1.58 ± 0.29 and the participants with sporadic PD had 1.82 ± 0.25 [139].

### 3.7. Normal Cardiac MIBG Uptake in Parkinson’s Disease

An abnormal result in cardiac MIBG uptake in individuals with parkinsonism can be a supporting criterion for diagnosing PD. However, a normal result of cardiac MIBG in individuals with suspicion of parkinsonian syndrome does not exclude the diagnosis of PD.

Tsujikawa et al. particularly evaluated individuals with normal results in a cardiac MIBG study. They found that patients with PD with normal or mildly low MIBG uptakes at the early stages of illness were predominantly females, with young disease onset, slow disease progression in motor dysfunctions, and preserved cognitive function [140]. It is worth mentioning that the authors did not perform genetic analysis in their study, which can lead to misinterpretations. One possible explanation is that the variety of anatomical distributions of Lewy bodies could influence the variety of MIBG uptakes among patients with PD [141].

An interesting fact related to normal MIBG scans is that this group of individuals apparently have more significant damage from synuclein pathology in motor function than in non-motor manifestations compared with those with decreased MIBG, even when this variable was adjusted. Also, individuals with PD and normal cardiac MIBG have a relatively low disease burden compared with those with abnormal MIBG [142]. Interestingly, the relationship between disease burden and abnormal cardiac MIBG uptake was shown to be independent of orthostatic blood pressure fluctuations, suggesting that extracranial cardiac markers might reflect disease burden and disease progression in the central nervous system [143].

Oh et al. investigated dopamine transporter activity of the corpus striatum and thalamus according to cardiac MIBG uptake in patients with PD. They found that individuals with PD with normal cardiac MIBG uptake had preserved dopamine reserve in the globus pallidus region [144]. In this way, the progression of the disease to the brainstem can explain cardiac sympathetic denervation. And dopaminergic loss in the globus pallidus may be associated with the complex circuitry in the midbrain.

### 3.8. Sleep Behavior Disorder, Cognition, and Dysphagia

Isolated rapid-eye-movement sleep behavior disorder (iRBD) is a prodromal stage of Lewy body disease and multiple system atrophy. Park et al. investigated plasma neurofilament light chain and cardiac MIBG uptake as predictors for phenoconversion. They found that elevated plasma neurofilament light chain levels may suggest imminent phenoconversion to multiple system atrophy, whereas low cardiac MIBG uptake suggests phenoconversion to Lewy body disease [145]. A possible explanation for the association between the plasma neurofilaments light chain levels and multiple system atrophy is the fact that neurodegeneration in this condition usually involves pontine nuclei leading to axonal degeneration in projecting fibers, which have large myelinated axons that are abundant in neurofilaments light chain [146].

Cognitive impairment in individuals with Lewy body pathology can be related to a cardiovascular autonomic dysfunction or the direct lesion of Lewy bodies in the neocortex [147]. Interestingly, in individuals with PD, there is a strong relationship between cognitive impairment and the degree of autonomic dysfunction in MIBG, which can suggest impaired microvascular circulation or invasion of α-synuclein in the central nervous system. Noteworthy, abnormal cognitive function is not associated with autonomic dysfunction in individuals with dementia with Lewy bodies, which may suggest initial involvement of Lewy body pathology in the neocortex [148].

Youn et al. found an association between cardiac sympathetic denervation and the presence and severity of dysphagia in patients with PD. The mean early and late H/M ratios were significantly lower in the PD with dysphagia group than the without dysphagia group (1.39 ± 0.21 vs. 1.86 ± 0.21, *p* < 0.01; 1.26 ± 0.18 vs. 1.82 ± 0.29, *p* < 0.01). The mechanism of dysphagia in PD is poorly understood, and these findings may suggest a cholinergic and noradrenergic dysfunction associated with striatonigral lesions [149].

### 3.9. Cardiac MIBG Scintigraphy in Pre-Motor Parkinson’s Disease

A relationship exists between motor symptoms and Lewy bodies and a-synuclein aggregation in the substantia nigra pars compacta. However, non-motor symptoms are believed to be independent of substantia nigra pathology, suggesting possible different mechanisms for the degenerative processes. Therefore, methods to predict the development of PD should be studied. In the following table, there is a list of symptoms that were already assessed with cardiac MIBG for the understanding and development of PD (Table 3).

## 4. Cardiac MIBG among Parkinsonian Syndromes

### 4.1. Dementia with Lewy Bodies

Cardiac autonomic denervation is observed in Lewy body disorders, including PD, dementia with Lewy bodies, and pure autonomic failure. The difference between PD and dementia with Lewy bodies is according to the main anatomical distribution of the Lewy body type-degeneration, which, for patients with PD, is in the basal ganglia and for individuals with dementia with Lewy bodies, is in the neocortical areas. In this context, cardiac MIBG uptake will be reduced in PD and dementia with Lewy bodies. Therefore, cardiac scintigraphy cannot differentiate these conditions [166]. Interestingly, the reduction in cardiac MIBG uptake in early and delayed H/M ratios is more significant in dementia with Lewy bodies than in patients with PD (1.47 ± 0.15 vs. 1.28 ± 0.13, *p* < 0.01; 1.78 ± 0.27 vs. 1.62 ± 0.38, *p* < 0.01) [167]. Moreover, a multicenter study on individuals with dementia with Lewy bodies found that the sensitivity and specificity of MIBG increased with the progression of the disease [168]. A similar effect on cardiac MIBG uptake was previously observed in patients with PD.

### 4.2. Multiple System Atrophy

Autonomic denervation occurs in PD and multiple system atrophy. There is severe and generalized preganglionic adrenergic involvement in multiple system atrophy. However, in PD, adrenergic lesions occur in pre- and postganglionic autonomic neurons. Therefore, cardiac MIBG uptake will be normal in multiple system atrophy and reduced in PD. The specificity of cardiac MIBG in differentiating PD from multiple system atrophy varies from 0.70 [73] to 0.95 [169] in the literature.

A significant overlap and different conditions can affect postganglionic adrenergic neurons, especially in the geriatric population. Skowronek et al. revealed many conditions affecting the resulting cardiac MIBG scintigraphy and misleading the diagnosis of multiple system atrophy. The authors found that 44% of the individuals with multiple system atrophy phenotypes did not correspond to the cardiac imaging. Common pitfalls were autonomic neuropathy, medication, structural cardiac disease, and erroneous calculation of the H/M ratio [110].

### 4.3. Corticobasal Degeneration

Autonomic dysfunction is usually not observed in patients with corticobasal degeneration. Also, dysautonomia and cerebellar dysfunction are among the exclusion criteria for sporadic corticobasal degeneration and possible corticobasal degeneration [170].

Orimo et al. assessed cardiac MIBG studies and [201Tl]-Cl myocardial scintigraphy in individuals with corticobasal degeneration, PD, and healthy participants. They observed that individuals with corticobasal degeneration presented with normal cardiac MIBG uptake and autonomic function tests compared with PD. Noteworthy, there was no significant difference in the cardiac and autonomic function studies on individuals with corticobasal degeneration compared with healthy volunteers [78]. Taki et al. and Kawazoe et al. found similar results when assessing cardiac MIBG scintigraphy in individuals with corticobasal degeneration [58,74].

### 4.4. Progressive Supranuclear Palsy

Predominant and unexplained autonomic failure are exclusion criteria for diagnosing progressive supranuclear palsy (PSP). In individuals with PSP with predominant parkinsonism, autonomic dysfunction is rarely encountered compared with patients with PD. The same should be applied to PSP with predominant corticobasal syndrome, in which severe autonomic signs or cerebellar disturbances may suggest multiple system atrophy. PSP and multiple system atrophy are the most likely causes of unexplained postural instability and falls. However, the cause of these neurological findings in multiple system atrophy is related to orthostatic hypotension and autonomic disturbances [171].

The literature has controversial results about progressive supranuclear palsy and cardiac MIBG. Some studies showed significantly reduced cardiac MIBG uptake in 85% of patients [80]. Further studies with better design and detailed criteria for PSP diagnosis revealed that 16% [74] to 21% [68] of patients presented abnormal cardiac MIBG uptake. However, the clinically different phenotypes of PSP and other movement disorders probably led to these different results. Furthermore, recent data demonstrated that individuals who fulfill the PSP criteria have normal cardiac MIBG uptake [58].

### 4.5. Essential Tremor

Clinical criteria are enough to diagnose most patients presenting with essential tremors. Imaging techniques can be used in essential tremors to differentiate from parkinsonism, mainly in long-term essential tremors presenting with equivocal signs of PD and with emerging signs of parkinsonism. Nevertheless, most studies were performed in individuals with a clear diagnosis of essential tremors, or when patients had the cardinal features of PD [172].

The clinical criteria for diagnosing essential tremors are isolated tremors, consisting of bilateral upper limb action (kinetic and postural) tremors, without other motor abnormalities. Therefore, the results of cardiac MIBG should be normal. Cardiac MIBG can effectively distinguish patients with early PD or tremor-dominant PD, which are sometimes challenging to differentiate from essential tremors [83].

## 5. Dual Imaging Algorithm—Central and Peripheral Imaging

The number of individuals with PD that are misdiagnosed during the first five years of the disease is 74% [173]. In this context, including supporting criteria such as cardiac MIBG may significantly decrease these misdiagnosis rates to less than ten percent. Some authors propose the inclusion of cardiac MIBG associated with neuroimaging techniques that evaluate dopamine transport, like [123I]-Ioflupane. PD’s central nervous system effects have been the main research concern for decades. However, there is significant evidence of disease progression to superior cervical sympathetic ganglia, mesenteric plexus of the GI tract, and widespread autonomic neuropathy [174]. Therefore, imaging evaluating the central nervous system associated with a peripheral nervous system assessment should be considered an option to investigate PD progression.

Yoshii et al. found that the combined use of [123I]-FP-CIT and cardiac MIBG scintigraphy increases the specificity of PD diagnosis [175]. The authors found that the sensitivity and specificity of diagnosing PD were 91.7% and 15.0% using [123I]-FP-CIT and 78.3% and 90.0% for cardiac MIBG. However, when both techniques were applied, the sensitivity and specificity were 74.2% and 95.0%. It is worth mentioning that I-ioflupane scintigraphy is only significant in combination with cardiac MIBG when the specific binding ratio of ioflupane is greater than 3.8 [176].

Some studies revealed a significant effect of aging on the results of dual imaging to predict Lewy body pathology [177]. In one study, a subgroup of patients had abnormal cardiac MIBG uptake and normal dopamine transporter imaging [178]. Nuvoli et al. showed that patients who present with only peripheral neuroimaging abnormality are probably in an early stage of PD and may show a progressive condition [179]. Janzen et al. observed that 75% of individuals with a sleep behavior disorder with similar neuroimaging findings developed PD [180].

Sakuramoto et al. studied the combination of the midbrain-to-pontine ratio and cardiac MIBG scintigraphy to differentiate PD from other parkinsonian syndromes [111]. The individual diagnostic sensitivity of the midbrain-to-pontine ratio for PD and multiple system atrophy was 87.1%, and for progressive supranuclear palsy, the ratio was 78.6%. However, including cardiac MIBG studies significantly improved the diagnostic specificity for PD and multiple system atrophy to 100% and for progressive supranuclear palsy to 90%.

Fujita et al. studied cardiac MIBG, olfactory testing, and transcranial ultrasound in distinguishing PD from other parkinsonian syndromes. The authors found that one abnormal result from three tests improved sensitivity (86.1%) but decreased specificity (63.2%). When the authors analyzed two abnormal results from the three tests, good discrimination for early-stage PD (50% sensitivity and 93% specificity) and overall discrimination between the parkinsonian syndromes (57% sensitivity and 95% specificity) was observed [105].

## 6. Automated Techniques in Cardiac MIBG Scintigraphy

The most helpful quantitative measurement in cardiac MIBG is the H/M ratio calculated in the early and delayed phases based on static planar images. The technique to obtain these values is drawing cardiac regions of interest manually. Noteworthy, the parameters of the cardiac region, such as position and size, and the level of experience in drawing planar images can affect the final measurement of the H/M ratio. Other factors that can influence the quality of the image are the amount of [123I]-MIBG injected, the timeframe when the image was obtained, the percentage of energy window centered, and the quantity of energy in the photopeak [181].

Boccalini et al. investigated the effect of manual and semi-automatic methods for assessing MIBG semi-quantitative indices. The agreement between raters’ classification of pathological and non-pathological was lower for the manual (0.45 in the early phase and 0.69 in the delayed phase) than the semi-automatic (0.78) method. Also, Cohen–Kappa values showed that the semi-automatic method improved the agreement between expert and inexpert raters from 0.29 to 0.90 in the early phase of manual to semi-automated methods [182].

The manual method is the most common method to obtain a cardiac region of interest. However, Veltman et al. published a guideline to obtain fixed-size regions of interest for standardizing cardiac MIBG scintigraphy. Veltman et al. showed good reproducibility of fixed-size mediastinal and cardiac regions of interest in assessing H/M ratios in patients with structural heart disease. Veltman’s technique involves the design of an “anatomical landmark square” formed by the lung apexes (upper square border), the upper cardiac border (lower square border), and the medial contours of the lungs (medial square borders) to place fixed-size mediastinum and oval cardiac regions of interest. Using PMOD software Version 3.6 (PMOD Technologies Ltd., Fällanden, Switzerland; accessed on 2011), the operator can move a fixed-size region of interest to an anatomical landmark square, which is semi-automatic [183].

There are significant differences in H/M ratios among different facilities when evaluating multicenter studies. A cross-calibration phantom method was developed to convert institutional H/M ratios into standard H/M ratios comparable to the most common medium energy collimators. The collimator type was classified into low-energy and medium-energy types and included intermediate types. The range of the H/M ratio chosen was 1.6–1.7 for discriminating between good and bad prognosis, and a threshold of 2.0–2.2 for discriminating between normal and abnormal [184].

Measuring extracardiac structures’ MIBG uptake can provide significant information for understanding the peripheral pathology of PD. Similar to Veltman’s technique for the heart, other organs are being studied with semi-automatic methods. Ebina et al. assessed MIBG uptake in the parotid and submandibular glands, performing a new method developed for quantitative semi-automatic head and chest analysis. It is worth mentioning that the authors did not specifically investigate reproducibility and raters’ agreement [185].

Another issue regarding cardiac MIBG uptake is the cut-off values of the H/M ratio, which widely vary among different institutions. Nuvoli et al. used two machine learning techniques to provide reliable and reproducible H/M ratio cut-off values. Support Vector Machine and Random Forest classifier were applied. The authors found that both automatic classifiers for computer-assisted diagnosis of parkinsonian syndromes permitted a determination of a better cut-off [186].

## 7. Systematic Reviews

There are few systematic reviews of cardiac MIBG scintigraphy in PD. These systematic reviews should be cautiously analyzed. The main concern regarding studies on cardiac MIBG scintigraphy is that every facility has its protocols regarding the equipment, dose of [123I]-MIBG, and further imaging programming.

Braune et al. investigated 246 individuals with PD. They found that a sensitivity of 89.7% and specificity of 94.6% differentiate PD from multiple system atrophy [169]. Treglia et al. performed the largest systematic review with a meta-analysis of the literature. They included 1076 individuals with PD and 896 with other parkinsonian syndromes. Sensitivity was 88% (95% CI, 86–90%) and specificity was 85% (95%, CI 81–88%). The area under the ROC curve was 0.93 [187]. Orimo et al. assessed 625 individuals with PD. The sensitivity and specificity in the early H/M ratio were 82.6% and 89.2%, and in the delayed H/M ratio were 89.7% and 82.6%, respectively. When PD was limited to the early stage (Hoehn–Yahr stage 1 or 2), the pooled sensitivity and specificity for the delayed H/M ratio were 94.1% and 80.2%, respectively [188].

## 8. Expert Recommendations

There are many pitfalls in the interpretation of cardiac MIBG scintigraphy results. The most common population studied is elderly individuals, which can significantly increase the number of errors due to several comorbidities and medications associated with the aging effect (Table 4).

A major drawback of cardiac MIBG imaging is that some medications can affect the results. A skin biopsy may be a possible exam to combine with cardiac imaging. Also, this procedure is not influenced by cardiac disease and does not require drug discontinuation. Giannoccaro et al. investigated the simultaneous performance of cardiac MIBG and skin biopsy in individuals with PD. They observed that both procedures were abnormal in 91% of the cases and that the concordance was 82%. Interestingly, autonomic dysfunctions were often simultaneously widespread at the cardiac and skin sympathetic branches [189]. In a second study performed by the same group, the authors analyzed cardiac MIBG and skin phosphorylated α-synuclein deposits. They found that the combination of these methods showed a sensitivity of 97.5% and a determined specificity of 100% and 92.3%, respectively, in distinguishing Lewy body disorders and multiple system atrophy [190].

Another common limitation of cardiac MIBG imaging is unknown cardiovascular diseases in individuals with PD. Nakae et al. assessed the lung uptake of 123I-MIBG in the cardiac prognosis of cardiac MIBG uptake. They observed that increased lung 123I-MIBG uptake is useful to differentiate between heart disease and Lewy body disease. Also, they observed a direct relationship with the degree of left ventricle diastolic function [191]. One possible explanation for these findings is that the left ventricle dysfunction increases pulmonary artery pressure.

Some authors assessed the efficacy of cardiac MIBG to provide supporting features for determining the use of deep brain stimulation therapy. Asahi et al. noted that mean left-right specific binding ratios of a DaTscan of less than 3.0 and a delayed H/M ratio of less than 1.7 suggest a severe disease with Hoehn–Yahr stage 3 or 4 for PD, which can be an appropriate indication for deep brain stimulation therapy [192]. These results can be used to provide objective data for the neurosurgical management of PD.

The importance of investigating the periphery in cases of PD can be especially significant when assessing cases of secondary parkinsonism. Navarro-Otano et al. showed that the use of cardiac MIBG and, to a lesser extent, olfactory testing could assist the differential diagnosis between vascular parkinsonism and PD in subjects in which the diagnosis remains uncertain despite 123I-FP-CIT SPECT imaging [193]. Dopamine abnormalities in vascular parkinsonism are probably an indirect finding of lesions in the thalamocortical drive.

Three-dimensional imaging using SPECT can provide a complete understanding of a cardiac region of interest. In addition to a better anatomical view, the significant lung uptake of planar images can be minimized using SPECT [194]. Therefore, SPECT should be prioritized, if available.

Cardiac MIBG studies should be reserved for cases of neurological findings that are atypical to PD. Ikeda et al. revealed a small benefit in the early stage of PD in addition to neurological findings and dopaminergic trial response. Also, they found no cost-effectiveness in performing cardiac scintigraphy in the early phase of PD in Japan [195]. Interestingly, Asayama reported that the acute levodopa challenge test could be an immediate diagnostic tool for PD with sensitivity and specificity comparable to cardiac MIBG uptake [196].

In addition to dual imaging, another approach is the performance of serial images. Follow-up sequential cardiac MIBG studies showed a great improvement in sensitivity from 72% to 89%. Interestingly, a change in subsequent imaging may suggest the threshold hypothesis for synucleinopathy according to Braak staging [197]. Some patients can present with high Unified Parkinson’s Disease Rating Scale (UPDRS) motor scores and normal cardiac MIBG. In the follow-up, special attention should be given to the washout rate and delayed phase cardiac MIBG uptake [198,199].

**Table 4 brainsci-13-01471-t004:** Expert recommendations for cardiac MIBG scintigraphy.

Recommendation	Reference
1. The medication list of the patient should be assessed. A clear observation should be performed for those drugs that influence the results of cardiac MIBG studies. A remarkable decrease in lung uptake in cardiac MIBG imaging may suggest the influence of drugs.	[24]
2. Patients with structural cardiac conditions should be assessed case-by-case. Moreover, those individuals on use of inotropic agents should be assessed because false negative errors can occur. It is suggested to use polar model to have an anatominal localization of the defect and compare with previous cardiologic studies.	[194]
3. Autonomic neuropathy should be investigated in individuals with risk factors for autonomic dysfunction before imaging is ordered. Noteworthy, some patients may have normal clinical findings, but their underlying conditions can lead to localized ganglioneuropathies.	[110]
4. The heart-to-mediastinum ratio (H/M ratio) cut-off can significantly change the clinometric parameters of cardiac MIBG scintigraphy. A restrictive value can exclude many individuals with abnormal results.	[186]
5. Cardiac MIBG scintigraphy should be used only as supporting criteria. No absolute conclusions should be obtained from this indirect test.	[6]
6. The use of software to specifically assess the region of interest is recommended. Erroneous positions of the region of interest and abnormal anatomic positions can particularly lead to abnormal cardiac MIBG scintigraphy results.	[183]
7. An electrocardiogram is recommended for some patients, especially the elderly. There is a significant association between aging and sick-sinus syndrome. It is worth mentioning that this syndrome at the early stages is usually asymptomatic.	[103]
8. Cardiac MIBG imaging may be helpful in differentiating idiopathic PD from cases of secondary parkinsonism (drug-induced parkinsonism and vascular parkinsonism). Cardiac imaging should especially be applied when neuroimaging ([123I]-FP-CIT SPECT imaging) results can be misleading.	[200]
9. Three-dimensional images, compared with planar, should be prioritized when available. They can provide better anatomical localization and a lower number of artifacts.	[194]
10. Cardiac MIBG uptake abnormality with normal dopamine transporter imaging may suggest an early stage of PD.	[179]
11. Follow-up sequential cardiac MIBG studies may improve the diagnostic accuracy of an abnormal result. It is recommended to pay special attention to the washout rate and delayed phase cardiac MIBG uptake, which are the first to change.	[197]
12. The results of cardiac MIBG uptake can be used to assess the risk of developing syncope in individuals with PD. Also, it could be used as an objective finding of disease burden in patients with PD.	[123]
13. In individuals with unknown cardiovascular structural disease, lung [123I]-MIBG scintigraphy can be used to assess the influence of the heart disease in the cardiac MIBG imaging.	[191]

## 9. Future Studies

Future studies should include protocols regarding when cardiac MIBG should be requested during the PD timeline. The development of specific parameters for its request is recommended. There are still many drawbacks to the request of this type of imaging, including high costs without health insurance assistance and geographic availability.

Another point of discussion is the differences in diagnosing PD with immediate, early, and delayed registration of cardiac MIBG. The significance of these studies is especially related to the side effects of the radionuclide and the time required to collect image data. There is a significant difference between minutes and hours in the acquisition of images.

There is missing information on selected populations such as African Americans, Native Americans, and Hispanics. Most studies included PD individuals but did not describe their other comorbidities. In this way, a significant lack of information regarding the influence of different comorbidities in imaging, mainly in individuals with diabetes mellitus without signs of peripheral neuropathy, is observed in the literature. The study of comorbidities can minimize possible confounding factors or provide correction methods for expanding this method to this subgroup of individuals.

The clinical manifestations of drug-induced parkinsonism usually overlap those of PD. Moreover, the most common approach in drug-induced parkinsonism is discontinuing medication, but in some individuals with active neuropsychiatric comorbidities, this choice is not feasible. Therefore, dual neuroimaging investigating the central and peripheral systems is important for understanding the pathophysiology of different types of parkinsonism.

Quantifiable biomarkers of disease progression for use in clinical trials for neurological diseases should be designed. The development of diagnostic criteria with a broader approach to the disease, as proposed by dual neuroimaging, can lead to the development of calculators that provide prognosis to the individuals affected by PD. In this way, cardiac MIBG scintigraphy may be more fitting as a secondary outcome or as part of a composite outcome, in which multiple endpoints are combined.

## 10. Conclusions

The significance of cardiac MIBG scintigraphy may help support the diagnosis of PD. There are still no standard protocols with specific machine or radiotracer characteristics, which probably influenced the results in the literature and can lead to significant statistical errors in the development of systematic reviews with meta-analysis. Pre-motor symptoms in individuals with PD may be associated with abnormal cardiac MIBG uptake and normal dopamine transporter imaging. Neuroimaging findings can be used to predict the development of PD. Furthermore, in individuals with PD, cardiac MIBG can provide evidence for patient-centered care for choosing the most specific medication with a lower number of side effects.

## Figures and Tables

**Figure 1 brainsci-13-01471-f001:**
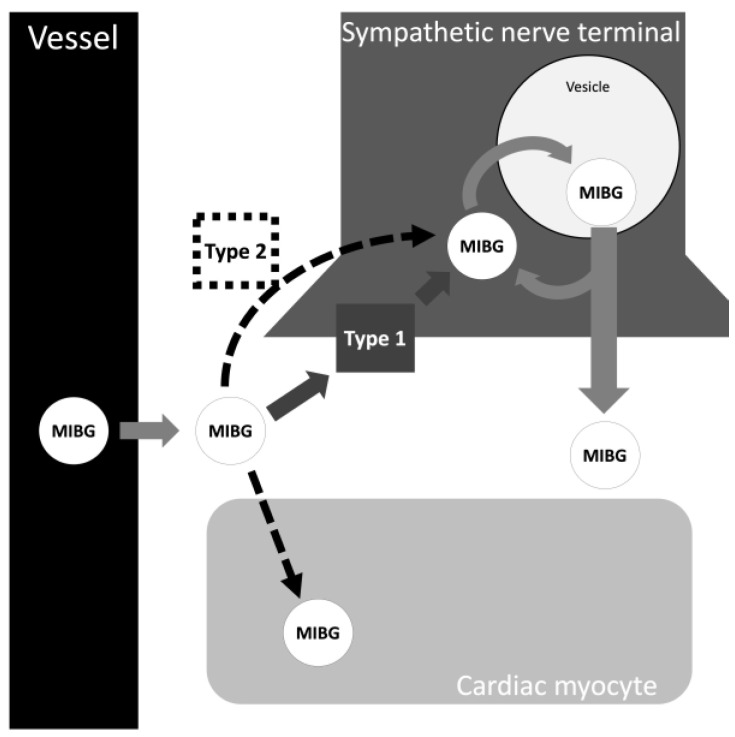
The two types of metaiodobenzylguanidine (MIGB) uptake into cells. Type 1 is an active pathway. Type 2 is a passive pathway by simple diffusion.

**Figure 2 brainsci-13-01471-f002:**
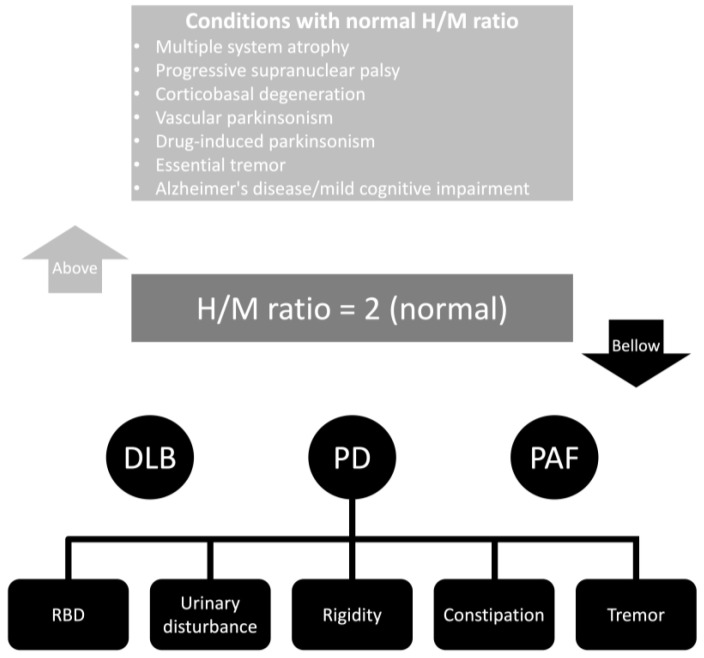
Cardiac MIBG results in different neurological conditions. Abbreviations: DLB, dementia with Lewy bodies; H/M, heart-to-mediastinum ratio; PAF, pure autonomic failure; PD, Parkinson’s disease; RBD, rapid eye movement sleep behavior disorder.

**Figure 3 brainsci-13-01471-f003:**
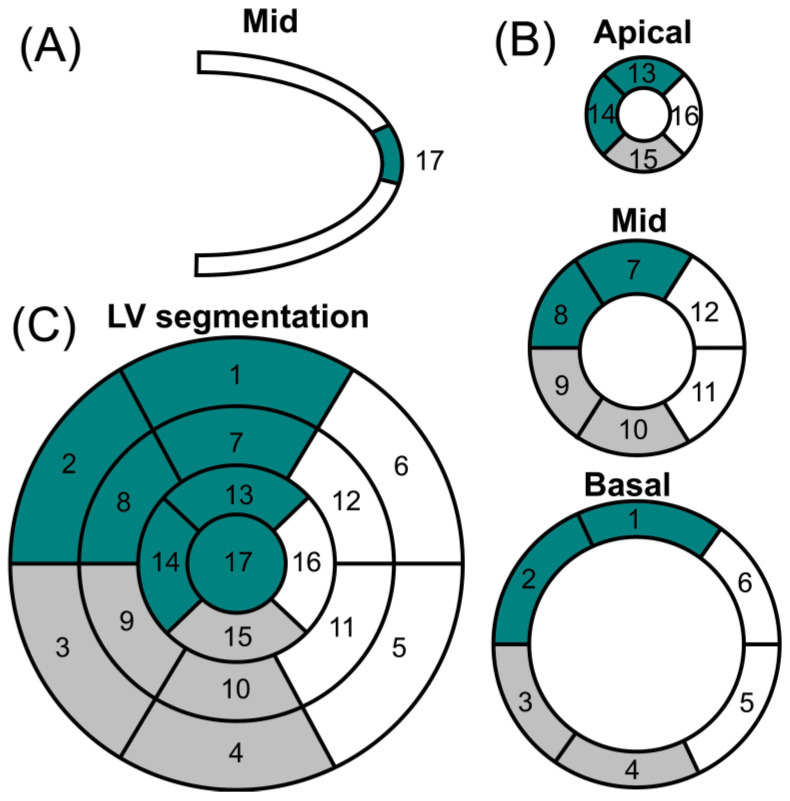
Polar map of cardiac perfusion. (**A**) Vertical long axis; (**B**) short axis; and (**C**) left ventricular (LV) segmentation. Coronary artery territories: left anterior descending artery (dark green); right coronary artery (gray); left circumflex artery (white). Segments: (1) basal anterior; (2) basal anteroseptal; (3) basal inferoseptal; (4) basal inferior; (5) basal inferolateral; (6) basal anterolateral; (7) mid anterior; (8) mid anteroseptal; (9) mid inferoseptal; (10) mid inferior; (11) mid inferolateral; (12) mid anterolateral; (13) apical anterior; (14) apical septal; (15) apical inferior; (16) apical lateral; (17) apex.

**Table 1 brainsci-13-01471-t001:** Interference from some drugs in the uptake of MIBG by Solanki et al. [20] modified by Rissardo et al.

Mechanism	Medication	Time to Hold before the Procedure
Inhibition of sodium-dependent uptake	Fluphenazine, pipotiazine, zuclopenthixol	1 month (depot)
	Labetalol, levomepromazine (methotrimeprazine), pimozide	72 h
	Amitriptyline, amoxapine, benperidol, butriptyline, flupenthixol, haloperidol, lofepramine, loxapine, maprotiline, mianserin, periciazine, trimipramine, thiethylperazine, trifluoperazine, trimipramine, viloxazine, zuclopenthixol	48 h
	Chlorpromazine, clomipramine, cocaine, desipramine, doxepin, dothiepin, droperidol, ephedrine, imipramine, iprindole, oxypertine, nortriptyline, perphenazine, prochlorperazine, promazine, promethazine, protriptyline, thioridazine, trifluperidol	24 h
Inhibition of uptake by active transport into vesicle	Methoserpidine, reserpine	72 h
	Xamoterol	48 h
Competition for transport into vesicle	Bethanidine, bretylium, debrisoquine, guanethidine	48 h
Depletion of content from storage vesicle	Labetalol, methoserpidine, reserpine	72 h
	Amfepramone (diethylpropion), bethanidine, bretylium, debrisoquine, dextroamphetamine, fenfluramine, guanethidine, mazindol, phentermine, phenylephedrine, phenylpropanolamine, pirbuterol, trazodone	48 h
	Dopamine, dobutamine, fenoterol, isoetharine, isoprenaline (isoproterenol), metaraminol, methylephedrine, methoxamine, norepinephrine (noradrenaline), orciprenaline, pseudoephedrine, reproterol, rimiterol, albuterol (salbutamol), terbutaline	24 h
Calcium-mediated	Isradipine, lidoflazine, nicardipine, verapamil	48 h
	Diltiazem, nifedipine, nimodipine	24 h

**Table 2 brainsci-13-01471-t002:** Sensitivity, specificity, and H/M ratio of cardiac MIBG scintigraphy in Parkinson’s disease.

Reference	n	n PD	H&Y Stage ^a^	H/M Ratio				Sensitivity ^c^		Specificity ^c^	
				PD ^b^		Cutoff		Early Phase	Delayed Phase	Early Phase	Delayed Phase
				Early Phase	Delayed Phase	Early Phase	Delayed Phase				
Yoshita et al. [68]	54	25	2.1	1.36 ± 0.15	1.19 ± 0.15	NA		1.00		0.79	
Braune et al. [69]	10	10	NA	NA	1.06 ± 0.06	NA		NA		NA	
Iwasa et al. [70]	12	12	2.5	1.55 ± 0.17	1.37 ± 0.15	NA		NA		NA	
Braune et al. [71]	20	15	NA	NA	1.08 ± 0.13	NA		NA		NA	
Orimo et al. [72]	68	45	3.0	1.71 ± 0.36	1.53 ± 0.36	NA		0.80		0.87	
Druschky et al. [73]	30	10	1.6	NA	1.25 ± 0.61	NA		NA		NA	
Taki et al. [74]	70	41	1.9	1.61 ± 0.29	1.47 ± 0.34	1.89	2.02	0.83	0.90	0.83	0.76
Reinhardt et al. [75]	28	21	3.4	1.05		NA		1.00		1.00	
Takatsu et al. [76]	32	32	2.84	1.58 ± 0.37	1.33 ± 0.28	NA		0.93		1.00	
Courbon et al. [61]	28	18	2.34	NA	1.50 ± 0.53	NA	1.30	0.80		1.00	
Hamada et al. [77]	113	88	2.61	1.51 ± 0.32	1.39 ± 0.33	NA		NA		NA	
Orimo et al. [78]	98	90	2.73	1.72 ± 0.33	1.54 ± 0.35	NA		NA		NA	
Saiki et al. [79]	45	34	2.55	1.45 ± 0.20	1.33 ± 0.27	1.38	1.25	0.83	0.66	0.86	0.73
Nagayama et al. [80]	391	122	3.0	NA	1.38 ± 0.29	NA	1.84	NA	0.87	NA	0.37
Kashihara et al. [81]	204	130	3.1	1.63 ± 0.29	1.37 ± 0.27	NA		0.84		1.00	
Kim et al. [82]	65	30	NA	NA	1.27 ± 0.13	NA		1.00		0.84	
Lee et al. [83]	93	51	1.61	NA	1.28 ± 0.35	NA		0.98		1.00	
Miyamoto et al. [84]	34	12	2.3	NA	1.43 ± 0.20	NA		NA		NA	
Shin et al. [85]	119	40	2.25	1.34 ± 0.15	1.29 ± 0.15	1.38	1.36	0.65	0.80	0.95	1.00
Köllensperger et al. [86]	18	9	3.2	1.51 ± 0.24	1.32 ± 0.25	1.93	1.68	0.44	0.55	0.88	0.88
Spiegel et al. [87]	102	102	1.7	NA	1.45 ± 0.29	NA		NA	0.93	NA	
Miyamoto et al. [88]	95	26	NA	2.08 ± 0.55	1.80 ± 0.68	1.82	NA	0.65	NA	0.77	NA
Chung et al. [89]	51	27	2.5	1.53 ± 0.27	1.35 ± 0.24	1.74	1.79	0.85	1.00	0.54	0.68
Novellino et al. [90]	70	20	NA	NA	1.10 ± 0.09	NA		1.0		1.0	
Sawada et al. [91]	400	267	3.2	1.66 ± 0.33	1.44 ± 0.39	1.92	1.68	0.81	0.84	0.85	0.89
Fröhlich et al. [92]	50	39	NA	NA	1.48 ± 0.46	NA	1.60	NA	0.87	NA	0.46
Ishibashi et al. [93]	39	24	2.41	1.66 ± 0.45	1.46 ± 0.41	1.95	1.60	0.79	0.93	0.70	0.93
Izawa et al. [94]	80	44	NA	1.67 ± 0.37	NA	1.66	NA	0.60	NA	NA	
Kikuchi et al. [95]	84	42	NA	NA	1.55 ± 0.30	NA	1.75	NA	0.85	NA	0.76
Kurata et al. [96]	295	166	2.96	1.74 ± 0.41	1.53 ± 0.48	NA		NA		NA	
Muxí et al. [97]	28	14	1.57	1.28 ± 0.11	1.12 ± 0.11	1.48	1.43	0.86	0.93	0.92	1.00
Südmeyer et al. [98]	48	31	NA	NA	1.34 ± 0.27	NA	1.34	NA	0.88	NA	0.65
Behnke et al. [99]	42	42	1.47	NA	1.47 ± 0.31	NA		NA		NA	
Chiaravalloti et al. [100]	37	37	1.67	1.72 ± 0.33	1.6 ± 0.32	NA		NA		NA	
Umemura et al. [101]	138	118	NA	NA	1.75 ± 0.63	NA	1.85	NA	0.67	NA	0.80
Leite et al. [102]	21	21	2	1.53 ± 0.27	1.46 ± 0.29	1.8	1.7	NA		NA	
Katagiri et al. [66]	100	50	2.3	2.05 ± 0.68	1.84 ± 0.88	NA		NA		NA	
Mochizuki et al. [103]	357	191	2.3	1.91 ± 0.51	1.62 ± 0.60	NA		NA		NA	
Rocchi et al. [104]	27	27	2.4	NA	1.53 ± 0.39	NA		NA		NA	0.70
Tsujikawa et al. [49]	70	70	2.1	1.83 ± 0.40	1.69 ± 0.48	1.90	1.97	1.00	0.64	1.00	0.71
Fujita et al. [105]	139	101	2.7	NA	1.9 ± 0.1	NA	2.00	NA	0.70	NA	0.86
Uyama et al. [106]	34	15	NA	NA	2.19 ± 0.55	NA	2.74	NA	0.86	NA	0.79
Yang et al. [107]	64	25	2	1.65 ± 0.36	1.50 ± 0.43	NA		NA		NA	
Gabilondo et al. [108]	194	85	NA	1.80 ± 0.51	1.60 ± 0.46	2.16	NA	0.87	NA	0.89	NA
Brandl et al. [109]	167	104	2.0	NA	1.26 ± 0.24	NA		NA	0.94	NA	0.65
Kawazoe et al. [58]	600	272	2.39	NA	NA	2.00	2.00	0.74	0.82	0.75	0.84
Skowronek et al. [110]	36	11	NA	NA	1.5 ± 0.5	NA		NA		NA	0.73
Jeong et al. [35]	60	60	2.2	1.39 ± 0.15	1.31 ± 0.15	NA		NA		NA	
Sakuramoto et al. [111]	96	70	2.6	NA	1.99 ± 0.89	NA	2.00	NA	0.67	NA	1.00
Brumberg et al. [112]	42	21	NA	NA	1.94 ± 0.63	NA	2.76	NA	0.90	NA	0.66
Jang et al. [113]	31	31	NA	2.10 ± 0.87	1.85 ± 1.22	NA		NA		NA	
Iwabuchi et al. [114]	216	90	NA	1.99 ± 0.44	1.82 ± 0.54	NA	2.26	NA		NA	
Eckhardt et al. [63]	31	19	2.42	NA	1.18 ± 0.19	NA		NA	0.89	NA	0.67
Miyagi et al. [115]	28	17	2.0	1.92 ± 0.56	1.69 ± 0.71	2.2	2.2	NA		NA	

Abbreviations: H/M, heart-to-mediastinum; H&Y, Hoehn and Yahr stages; N, number of participants; NA, not available/not applicable; PD, Parkinson’s disease. ^a^ The mean scores of Hoehn and Yahr stages; ^b^ the mean scores and standard deviation of H/M ratio; ^c^ the sensitivity and specificity were described according to early and delayed phase registration. When only a number is provided, the results represent the authors’ description of general data.

**Table 3 brainsci-13-01471-t003:** Cardiac MIBG scintigraphy in pre-motor Parkinson’s disease.

Symptom	Description	Differential Diagnosis	Reference
Constipation	Constipation and abnormal cardiac MIBG were associated with the initial manifestation of PD. One in every four individuals with this presentation will have a diagnosis of PD.	DLB	[22,150]
Postural hypotension	Pure autonomic failure may be a risk factor for the development of PD. There is a significant association between reduced cardiac MIBG in patients with pure autonomic failure and the development of PD.	MSA	[81,151]
Cognitive impairment	Most patients who present with cognitive impairment and abnormal cardiac MIBG will develop dementia with Lewy bodies; only a small percentage of individuals will develop PD. Reduced cardiac MIBG may be associated with a subsequent risk of dementia and may reflect the wider extension of alpha-synuclein pathology.	DLB	[152,153]
Depression/ anxiety	Depression is among the most common neuropsychiatric disorders in PD. Of 13 patients presenting with depression and anxiety and abnormal cardiac MIBG, 11 developed PD years later.	DLB	[154,155]
Visual hallucinations	Visual hallucinations are a possible risk factor for the development of PD. Four individuals presented with visual hallucinations and abnormal cardiac MIBG, but no discussion is given in the follow-up of these individuals. Also, visual hallucinations is an independent risk factor for abnormal cardiac MIBG uptake.	Corticobasal degeneration	[154,156]
Sleep disorders	Low MIBG uptake and rapid eye movement sleep behavior disorder was associated with PD severity. Also, REM sleep behavior disorder is an independent risk factor associated with abnormal MIBG uptake in individuals with PD.	DLBMSA	[157,158,159]
Olfactory	A relationship between abnormal MIBG uptake and hyposmia in the early prodromal stage of PD (before nigrostriatal degeneration) was observed. Degeneration in broad aspects of the cardiovascular sympathetic system occurs concurrently with olfactory system degeneration during the premotor phase of PD.	DLB	[160,161]
Gait	Abnormal MIBG uptake was already associated with a higher incidence of falls. This may be an indirect finding of autonomic dysfunction. Also, cardiac MIBG uptake correlated significantly with the annual progress of rigidity and axial symptoms.	MSA	[162,163]
Fatigue	The results in the literature are contradictory. Some authors found, and others did not, a correlation between cardiac MIBG uptake and PD-related fatigue.	MSA	[164,165]

Abbreviations: DLB, dementia with Lewy bodies; MSA, multiple system atrophy; PD, Parkinson’s disease.

## Data Availability

Not applicable.

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
