# Peer review of "Cardiac 123I-Metaiodobenzylguanidine (MIBG) Scintigraphy in Parkinson’s Disease: A Comprehensive Review"

_brainsci, 2023, doi:10.3390/brainsci13101471_

Round 1

Reviewer 1 Report

In this manuscript, the authors aimed to provide a comprehensive review of the MIBG cardiac scan and its diagnostic utility in Parkinson's disease. The manuscript is very well written and the data provided are in-depth. I have only a few suggestions;

It is better to add the section about the association of cardiac MIBG with motor symptoms such as freezing of gait, and postural instability. Also, about the association between MIBG scintigraphy and LEDD (levodopa equivalent daily dose) and types of anti-parkinsonian medications.

Author Response

In this manuscript, the authors aimed to provide a comprehensive review of the MIBG cardiac scan and its diagnostic utility in Parkinson's disease. The manuscript is very well written and the data provided are in-depth. I have only a few suggestions; It is better to add the section about the association of cardiac MIBG with motor symptoms such as freezing of gait, and postural instability. Also, about the association between MIBG scintigraphy and LEDD (levodopa equivalent daily dose) and types of anti-parkinsonian medications.

Authors:

Cardiac MIBG and some clinical characteristics of PD are described in Table 3. The gait was analyzed in the literature with the number of falls, but no specific description of the freezing characteristic was provided. Also, the postural instability was analyzed as a postural hypotension.

The authors re-reviewed the literature regarding LEDD and other medications with cardiac MIBG. Matsui et al. (2006) found no significant statistical correlation between LEDD and cardiac MIBG. Also, the current antiparkinsonian medication did not interfere with the uptake of MIBG (Kishi et al. 2011). For a complete description of the list of the medications that could influence MIBG, Table 1 is provided.

Reviewer 2 Report

Manuscript ID: brainsci-2608370

Cardiac 123I-Metaiodobenzylguanidine (MIBG) Scintigraphy in

Parkinson’s Disease: A Comprehensive Review

Many influencing factors, particularly comorbidities and medications  on the results of cardiac MIBG spect were not considered.

There is not much new information in this review

9/23

moderate

Author Response

Manuscript ID: brainsci-2608370

Cardiac 123I-Metaiodobenzylguanidine (MIBG) Scintigraphy in

Parkinson’s Disease: A Comprehensive Review

Many influencing factors, particularly comorbidities and medications  on the results of cardiac MIBG spect were not considered.

There is not much new information in this review

9/23

Authors:

We appreciate the opinion of the Reviewer regarding our manuscript. In the present manuscript, it was reviewed all the literature regarding cardiac MIBG and Parkinson’s disease. The technique description provides insightful comments regarding some tips for acquiring the images. Another chapter describes the cardiac MIBG and its correlations with the pathophysiology of Parkinson’s disease. Followed by describing some risk factors of Parkinson’s disease and their evaluation of cardiac MIBG to assess the risk of developing Parkinson’s disease. The other chapters about dual imaging algorithms and automated techniques are worthwhile for future research. In the expert recommendation section, we did a table to provide some tips on using cardiac MIBG.

To the authors' knowledge, there is no compared study in the literature with these descriptions.

Reviewer 3 Report

Dear authors, the review is very interesting and its structure is appropriate. However, there are only a few points to resolve.

1) Table 2, the data recorded from the reviewed articles could perform some statistical tests such as contingency tables. This would allow for a more solid framework to reach more robust conclusions.

2) Review the format of the references, some are not homogenized with the guidelines.

3) Adjust the margin line of Table 1 (it joins line 106).

4) A minor adjustment to the language style would be pertinent.

Minor editing of English language required

Author Response

Dear authors, the review is very interesting and its structure is appropriate. However, there are only a few points to resolve.

1) Table 2, the data recorded from the reviewed articles could perform some statistical tests such as contingency tables. This would allow for a more solid framework to reach more robust conclusions.

Authors: The authors believe that the data studied does not allow a contingency table. The only data with a total is the number of individuals in the study and the number of individuals with PD that we believe in the present format is better visualized.

2) Review the format of the references, some are not homogenized with the guidelines.

Authors: We appreciate the reviewer's comment. The references were reviewed and organized according to the author’s guidelines.

3) Adjust the margin line of Table 1 (it joins line 106).

Authors: The margin line of Table 1 was adjusted.

4) A minor adjustment to the language style would be pertinent.

Authors: The authors reviewed the manuscript to address this comment.